# Change in Primary (Cr, Fe)_7_C_3_ Carbides Induced by Electric Current Pulse Modification of Hypereutectic High Chromium Cast Iron Melt

**DOI:** 10.3390/ma12010032

**Published:** 2018-12-22

**Authors:** Baoyu Geng, Rongfeng Zhou, Lu Li, Haiyang Lv, Yongkun Li, Dan Bai, Yehua Jiang

**Affiliations:** 1Faculty of Materials Science and Engineering, Kunming University of Science and Technology, Kunming 650093, China; gengbaoyu@kmust.edu.cn (B.G.); lilukust@126.com (L.L.); lvhaiyang@cqwu.edu.cn (H.L.); liyongkun@kmust.edu.cn (Y.L.); baidankmust@163.com (D.B.); jiangyehua@kmust.edu.cn (Y.J.); 2Research Center for Analysis and Measurement, Kunming University of Science and Technology, Kunming 650039, China; 3School of Materials and Chemical Engineering, Chongqing University of Arts and Sciences, Chongqing 402160, China

**Keywords:** (Cr, Fe)_7_C_3_, electric current pulse, micro defect, micro hardness

## Abstract

In this work, an electric current pulse (ECP) of 500A was applied on a hypereutectic high chromium cast iron (HHCCI) melt before it began to solidify, and the effect of ECP on primary carbides was investigated. The characteristics of the primary carbides were analyzed by X-ray diffraction (XRD), electron probe micro-analyzer (EPMA), transmission electron microscopy (TEM), micro hardness tester, and other techniques. The results showed that ECP not only refined the primary (Cr, Fe)_7_C_3_ carbides, but also decreased the average content of Cr in the primary carbides. At the same time, the average value of micro hardness of the primary carbides increased by about 84 Kgf/mm^2^, which contradicts existing knowledge that hardness increases with an increase in Cr content. XRD analysis showed that the crystal structure of the primary carbides did not change. The results of EPMA indicated that the Cr/Fe ratio gradually decreased from the center to the edges of the carbide particles. Further investigation revealed that the uneven distribution of elements caused by ECP led to an increase in defects (including twins, antiphase boundaries, and dislocations). This increase in defect density is the main reason for the increase in micro hardness instead of the expected decrease. The mechanism of the change in primary carbides was analyzed in detail in this paper, which has provided a new method for the refinement of primary carbides and for improving the properties of primary carbides.

## 1. Introduction

M7C3 carbides are the main strengthening phase in hypereutectic high chromium cast iron (HHCCI) due to their greater strength and hardness (where M is mainly Fe and Cr, and C is carbon) [1,2]. The morphology, distribution, composition, and internal micro-defects of the primary carbides have important effects on the properties of the alloy [1,3,4]. Investigation of carbide characteristics and corresponding influencing factors is beneficial to improving the properties of alloys.

M7C3 carbides have been widely studied over the past century, which has provided a good foundation of knowledge for their further development. Regarding the morphology of carbides, it has been reported that primary carbides grow as rods while eutectic carbides grow as blades in longitudinal sections and as fine rods in transverse sections [5,6,7]. With regard to the structure of carbide crystals, three crystal structures have been proposed; these are the trigonal symmetry structure proposed by Westgren (1935) [8], the hexagonal crystal system proposed by Herbstein and Snyman (1964) [9], and the orthorhombic crystal system proposed by Rouault (1970) [10]. It has been pointed out that there are defects such as twins, antiphase boundaries, and dislocations in M7C3 carbides [11,12,13,14,15]. Regarding the performance of carbides, it has been found that the hardness of carbides increases with an increase in Cr content and that toughness follows the opposite trend [6,16]. It has been proven by theoretical calculation that an increase in Cr content is helpful to improving the hardness of carbides ((Cr, Fe)_7_C_3_) [17,18]. Studies have shown that the hardness and toughness of carbides in different sections (longitudinal sections and transverse sections) are anisotropic [6]. Previous reports [19] have suggested that the addition of alloy elements such as Mo and Ni can change the properties of carbides, but this undoubtedly increases the cost. Although much research has been performed on the characteristics of M7C3 carbides, the methods and mechanisms for improving carbide properties are still a focus in the field of ferrous metal materials.

Electric current pulse (ECP) modification is a suitable method for improving the microstructure and properties of materials because it is highly efficient, clean, and inexpensive. ECP processing has been studied and applied widely in many fields since it was first developed. Some of its applications include improving the plastic and fatigue properties of materials, the phase transition of solid metals, and controlling the solidification process of alloys [20,21,22,23,24,25,26]. Especially in the last 30 years, the application of ECP in solidification control processes has become a research focus. However, scholars have different explanations for the mechanism of grain refinement caused by ECP. These theories can be divided into two classes according to the temperature interval in which ECP is applied. The first theory focuses on the application of ECP to the solidification interval, and it mainly involves crystallization rain theory and dendrite breaking theory [22,23,27]. The second type focuses on the application of ECP to the melt interval and is based on the inoculation theory proposed by Wang et al. [28,29,30]. They have pointed out that ECP can distort the external electron layer of the crystal embryo (cluster) in pure metal melts, and that this distortion facilitates the refinement of grains [30]. Our previous studies have shown that ECP can effectively refine primary and eutectic carbides [31]. Based on our previous research, the present work aims to study the effects of ECP on the morphology, components, micro defects, and micro hardness of primary carbides in HHCCI. The mechanism of induction caused by ECP is also investigated.

## 2. Materials and Methods

In this work, the experimental material, HHCCI, contained 19.40% Cr, 3.84% C, 2.70% Si and balance Fe by weight. The start and end temperatures of melting were 1276 °C and 1337 °C, respectively, which were obtained using a simultaneous thermal analyzer (NETZSCH STA 449F3, NETZSCH, Selb, Germany). The experimental preformed samples (Φ18 mm × 150 mm) were first prepared by machining. The preformed samples were then sealed in an alundum tube by sodium silicate bonded sand and two nickel electrodes. The experimental device is shown in Figure 1. The preformed samples were heated for 160 min to raise the temperature to 1355 °C in a muffle furnace (YFK60_600/160, Shanghai Yifeng Electric Furnace Company, Shanghai, China), and then soaked at 1355 °C for 3 min. ECP of 500 A (500 A is the peak value of ECP) was passed through the melt between 1355 °C and 1337 °C by a sharp ECP generator. Subsequently, the melt solidified and was allowed to cool down to room temperature in the muffle furnace. According to our previous study, the optimal ECP parameters for grain refinement were 500 A, 45 Hz, and 10 μs [32]. Therefore, this set of parameters was used in this experiment. The temperature history of the melt was monitored by thermocouples. In the following sections, the sample treated by ECP is referred to as the ECP sample. The same heating, soaking, and cooling methods were used to obtain a comparative sample without pulsed current. This sample is referred to as the non-ECP sample in the following sections.

For optical microscope analyses, the samples were etched with 5% FeCl_3_ aqueous solution (5 g FeCl_3_ and 95 mL H_2_O). The morphology of samples was observed with a Leica optical microscope (Leica, Buffalo Grove, IL, USA). The micro hardness of the transverse sections of the primary carbides was measured with a HVS-1000A hardness tester (LaizhouWeiyi Experiment Machine Company, Lanzhou, China) (with a load of 300 gf and indentation time of 10 s, according to the ASTM E384 standard). The samples were extracted using 45% HCl aqueous solution (45% HCl and 55% H_2_O) over 120 h, and only primary carbides were selected manually. In order to speed up the extraction process, samples were wire-cut into 2 mm sheets and placed in 45% HCl aqueous solution. Then, ultrasonic waves were used to oscillate the HCl aqueous solution and the samples for 10 min every 24 h during extraction. Carbide particles were manually ground into powders for 0.5 min in an agate grinding dish at room temperature. The particle size after grinding was about 30–60 μm. X-ray diffraction (XRD) analysis of the powders was carried out with a D/max-3B diffractometer (Rigaku, Tokyo, Japan). XRD with Cu Kα radiation was used to identify the crystal phase and calculate the lattice parameters at a scanning speed of 0.013°/s and a 2θ range of 35° to 85°. In order to calculate the lattice parameters accurately, a standard silicon was used to calibrate 2θ of the XRD data. Morphology analysis of carbide powders was carried out by scanning electron microscopy (SEM) with a Nova Nano SEM 450 (FEI, Hillsboro, OR, USA). The content and distribution of elements on the transverse sections of the primary carbides were analyzed by electron probe micro-analyzer (EPMA) with a JXA8230 (JEOL, Tokyo, Japan). The area of the electron beam was approximately 1 μm^2^. The relative measurement error of the main elements (Fe, Cr) was less than or equal to 2%, and that of C was less than or equal to 6%. Micro defects in the primary carbides were observed by transmission electron microscopy (TEM) with a Tecnai G2 F30 (operated at an acceleration voltage of 300 KV) (FEI, Hillsboro, OR, USA). The TEM samples were prepared by focused ion beam (FIB) slicing technique with a TESCAN LYRA3 (TESCAN, Brno, Czech Republic).

## 3. Results

### 3.1. Morphology and Micro Hardness of the Primary Carbides

Figure 2 shows the morphology of the primary carbides in the non-ECP sample and the ECP sample. It was observed that the primary carbides were all hexagonal rods. Compared to the non-ECP sample, the primary carbides of complete hexagonal sections were significantly greater in the ECP sample. The size of the transverse sections of the primary carbides was reduced from about 200 μm to 100 μm. Moreover, the size of the longitudinal sections of the primary carbides was reduced from about 800 μm to 200 μm. Thus, the primary carbides were significantly refined by ECP. It should be mentioned that there was no obvious orientation phenomenon in the solidification structure. Therefore, it was difficult to accurately measure the size of the longitudinal directions of the primary carbides. In this experiment, grain size could only be evaluated by measuring the approximate size of transverse and longitudinal sections within the same section.

The micro hardness of five primary carbide particles in each sample were measured from transverse sections and the average values were obtained. Figure 3 shows indentation marks on transverse sections of the primary carbides. The average micro hardness value was 1409 ± 44 Kgf/mm^2^ for the non-ECP sample. It was 1493 ± 32 Kgf/mm^2^ for the ECP sample. These results indicate that the micro hardness of the primary carbides was improved slightly after ECP modification of the HHCCI melt.

### 3.2. Crystal Structure and Lattice Parameters of Primary Carbides

XRD analysis showed that the crystal structures of the primary carbides in both samples were not different and that they were M7C3 carbides (Figure 4). They had the orthorhombic crystal structure proposed by Rouault. However, compared to the non-ECP sample (a = 6.965 Å, b = 12.030 Å and c = 4.500 Å), the lattice parameters of primary carbides in the ECP sample (a = 6.960 Å, b = 12.018 Å and c = 4.495 Å) were smaller by 0.005, 0.012, and 0.005, respectively.

In addition, the diffraction peak intensities of some crystal planes in the ECP sample were larger than those in the non-ECP sample, including for the (150), (060), (260), (080), (550), and (390) planes.

### 3.3. Content and Distribution of Elements in Primary Carbides

The contents of elements in the transverse sections of primary carbides were analyzed by EPMA. Compared with the non-ECP sample, the average Cr content of primary carbides in the ECP sample was lower and the average Fe content was higher (Table 1). Considering the relatively large measurement error of C content, it can be assumed that both samples had equal C content. The calculated results showed that the (Fe, Cr)/C ratio of primary carbides was 7/3.3 for the non-ECP sample, and 7/3.2 for the ECP sample. The ratios of elements in the primary carbides of the two samples agreed with those of the M7C3 carbides. This was consistent with the XRD results.

Figure 5a,d shows the EPMA positions of the two samples using SEM imaging with backscattering electrons. Figure 5b–e shows the changes in elemental contents in the two samples, respectively. Figure 5c,f shows the trend of the atomic ratios of Cr/Fe. It can be seen that the Cr/Fe atomic ratios were almost constant in the central region of the primary carbides and then decreased suddenly at the edges for the non-ECP sample (Figure 5c). However, the ratios gradually decreased from the center to the edges for the ECP sample (Figure 5f). In other words, Cr content gradually decreased and Fe content gradually increased from the center to the edges.

### 3.4. Density of Micro Defects in Primary Carbides

The selected area diffraction patterns (SADP) from [001] zone axis of the primary carbides in both samples were similar (Figure 6b,e). According to analysis of the SADP, it could be concluded that they were M7C3 carbides. However, it can be clearly seen from the bright-field TEM micrographs that there were more dark stripes with regular arrangements in the carbides of the ECP sample than the non-ECP sample (the yellow arrow in Figure 6a,d). The angle of these dark stripes was 60° or 120°. The dark stripes were twins and antiphase boundaries [11]. It is important to note that the densities of the twins and antiphase boundaries in this work were lower compared to other literature data [11,12,13,14,15].

High resolution TEM (HRTEM) images of the carbides in the two samples obtained from the [001] zone axis are shown in Figure 6c,f. It can be seen that the bright spots in the HRTEM image along the red lines 1 and 2 have a regular arrangement from left to right in Figure 6c. However, there is a deviation in the arrangement of the bright spots along red lines 3 and 4 from left to right in Figure 6f. The left ends of red lines 3 and 4 were on bright spots, while the right ends were in the gap between the bright spots. Although no information concerning the detailed atomic arrangements can be obtained from the HRTEM image, HRTEM provides a powerful means for the direct observation of irregularities. Thus, it was observed that the lattice arrangements were regular at the left and right ends of the red lines 3 and 4, but the middle region (i.e., the region between the yellow dashed curves) was irregular. This irregular region was similar to a mixed dislocation with a width of 2–10 nm (Figure 6g). Other researchers have found dislocations in primary carbides [33]. By analyzing the bright-field TEM micrographs and HRTEM images, it can be seen that not only substructures (the twins or antiphase boundaries) but also dislocations within the substructures increased in the primary carbides of the ECP sample.

## 4. Discussion

### 4.1. Refinement of Primary Carbides Induced by ECP Melt Modification

ECP can distort the external electron layer of the crystal embryo (cluster) in the melt (Figure 7). This distortion facilitates the combination of free atoms in the melt with the crystal embryo. In addition, according to electromigration efficiency, metal ions will gain additional energy under the action of an electric field. The additional energy can reduce the activation energy for diffusion across a solid-liquid interface. These two factors can lead to an increase in the size of the crystal embryo. Larger crystal embryos are conducive to the subsequent formation of primary carbide nuclei and to the refinement of primary carbide grains.

It is well known that the solidification process of alloys is different from that of pure metals. Different types of ions will obtain different energies under the action of the same electric field. According to the free electron theory of metals, the number of valence electrons of Cr is 6 (3d^5^ 4s^1^), and that of Fe is 8 (3d^6^ 4s^2^). The electric quantity of a Fe ion core is greater than that of a Cr ion core when the current passes through the melt. However, their relative atomic masses are about the same (A_Cr_ = 52.00 g/mol, A_Fe_ = 55.85 g/mol). Due to the difference in their specific charges (specific charge being q/m, where q is charge and m is mass), the energy obtained by a Fe ion core is greater than that obtained by a Cr ion core during the same period of time under the same electric field. Fe ion cores with high energy combine more easily with crystal embryos in the nucleation stage (Figure 7b), which results in decreased content of Cr in crystal embryos. As the content of Cr in each crystal embryo decreases, the crystal nucleus of primary carbides increases when the Cr content of the alloy is constant. This is another reason for the increase in crystal nuclei of primary carbides caused by ECP.

Figure 7a shows the growth of crystal embryos without an electric field. The crystal embryo tends to receive Cr ion cores at the solid-liquid interface because of the higher cohesive energies of Cr-C. Figure 7b indicates growth under an electric field. In this case, the crystal embryo tends to receive more Fe ion cores because the Fe ion cores have higher energy to overcome the binding barrier.

The experiment in this study did not have the physical conditions required for crystallization rain theory and dendrite breaking theory because ECP was stopped at the melting point in this experiment. In addition, according to the slope of the fitting line in Figure 8, it can be seen that the cooling rate of the ECP sample was 0.12 °C/s and that of the non-ECP sample was 0.11 °C/s. The cooling rates of the two samples were quite similar and this slight difference can be neglected. Therefore, electromagnetic stirring did not cause accelerated loss of melt heat and it did not contribute significantly to the increase in nucleus quantity, possibly because the cooling mode was in-furnace cooling. It is worth noting that there were no characteristic stops in the cooling curve. The temperature range of the primary carbides precipitation reaction was large and the cooling rate was very small in this experiment (the cooling rate of melt is 1~100 °C/s for mold casting; however, it was 0.11 °C/s or 0.12 °C/s in this experiment), which were conducive to the release of latent heat of crystallization. This may be the reason for why there were no characteristic stops in the cooling curve. Solidification reaction only caused a slight decrease in the cooling rate (Figure 8).

Overall, the distortion of the external electron layer of crystal embryos and the differences in energy between Fe ions and Cr ions caused by ECP both contributed to the increase in the number of crystal nuclei during the initial stage of solidification. The increase in the number of primary carbide nuclei led to the refinement of the primary carbides.

### 4.2. Effect of ECP on Content and Distribution of Cr in Primary Carbides

Cr content in primary carbide nuclei is much higher than that in the liquid phase. Carbide nuclei with higher Cr content tend to receive more Cr atoms at the solid-liquid interface. Long-range diffusion of Cr is needed when the nucleus continues to grow. Therefore, Cr content in carbides is affected by long-range diffusion of Cr.

The ECP sample had more crystal nuclei than the non-ECP sample during solidification. Hence, Cr in the liquid phase was less than in the non-ECP samples during the initial stage of solidification. A lower concentration of Cr caused an insufficient diffusion driving force of Cr. This in turn was responsible for the lowering of Cr content at the solid-liquid interface. As the carbides grew, more and more Fe was received at the solid-liquid interface. This ultimately caused the Cr content to decrease uniformly from the center to the edges of the carbides (Figure 5e,f). However, the concentration of Cr in the liquid phase of the non-ECP samples was higher. The larger driving force of diffusion provided a continuous supply of Cr for the equilibrium growth of carbides until near-eutectic temperature. In the later stages of growth of the primary carbides, Cr could only diffuse over a short range due to the lower concentration of Cr. More Fe atoms were received at the end of the growth. This resulted in a uniform distribution of Cr in the central region of the primary carbides and a decrease in Cr content at the edges (Figure 5b,c). The schematic of this process is shown in Figure 9.

It is evident that average Cr content in the carbides of the ECP sample was lower, which was due to the influenced during two stages. First, ECP resulted in lower Cr content in the crystal embryo during the nucleation stage. Second, although ECP was removed during the growth stage, Cr content in the carbides was not restored to higher levels because the driving force of long-range diffusion of Cr was insufficient.

### 4.3. Decrease in Lattice Parameters of Primary Carbides

Compared with the non-ECP sample carbides, the average content of Fe was higher, Cr content was lower, and C content was almost equal in the ECP sample carbides (Table 1). The radius of a Fe atom is smaller than that of a Cr atom. Therefore, an increase in Fe will inevitably lead to a decrease in the lattice parameters of carbides, which is consistent with the XRD results.

It should be noted that the degrees of grinding for the two kinds of carbides were different because the original size of the carbides in the two samples was different. There were also rod-shaped carbides in the ECP sample and the side face of carbide rods can be seen (the red arrow in Figure 4b). However, the side faces of carbide rods were rarely seen in the non-ECP sample. This orientation exerted some influence on the XRD pattern. The direction along the carbide rods was the (001) direction of crystal [11]. The side face was parallel to <001> orientations (i.e., the index *l* of the crystal planes was zero). The diffraction opportunities of the side face would thus increase when the rod-shaped carbides existed in powdered form, for the ECP sample. Therefore, the intensity of the diffraction peaks of these crystal planes would increase. In other words, the relative variation of the peak intensities was due to the preferential distribution of the carbides.

### 4.4. Increase in Density of Defects and Micro Hardness of Primary Carbides

Error stacking or omission of C atoms are considered to be the causes of twins and antiphase boundaries [11,12,13,14]. There are two possible factors that can cause error stacking or omission of C atoms. The first factor is the uneven distribution of Fe and Cr elements. A gradient distribution of Cr and Fe leads to lattice distortion, which can give the C atom more chances to stand in the wrong position or to be omitted. The second possible factor is a higher cooling rate (we are not sure about this factor because we have not done the relevant verification in this work but have inferred this from relevant literature [11,12,13,14]). In another study, it has been found that twins and antiphase boundaries increase with an increase in cooling rates [11]. Preparation processes in that work were normal casting, ion plating, and sputtering. The cooling method for the two samples in our work was furnace cooling, and the cooling rate was lower than with normal casting conditions. The density of twins and antiphase boundaries observed in this work were lower than those reported in previous studies. The increase in twins and antiphase boundaries in the primary carbides of the ECP sample was mainly due to the uneven distribution of Cr and Fe elements in this experiment, because the cooling rate of the two samples was almost the same.

The Cr/Fe ratio gradually decreased from the center to the edges of the carbides, which led to an increase in dislocations in the substructures. An uneven Cr/Fe ratio caused differences in the internal and external lattice parameters. In other words, the lattice parameters became smaller from the center to the edges of the carbides. The lattice distortion stress would have been caused by the difference in lattice parameters inside and outside the carbides. In order to reduce this distortion energy, mixed dislocations were generated on (001). Several previous studies have indicated that the micro hardness of (Fe, Cr)_7_C_3_ increases with an increase in Cr content [6,16]. However, the micro hardness of the carbides in the ECP sample (with lower Cr content) was higher than that of the carbides in the non-ECP sample (with higher Cr content) in this work. It is well known that dislocation (or twins) strengthening is one of the main ways of strengthening. The hardness was higher mainly because there was a large number of boundaries and entanglement of mixed dislocation in the carbides.

## 5. Conclusions

(1)Compared to the primary carbides in the non-ECP sample, the grain sizes of the carbides in the ECP sample along the transverse and longitudinal directions reduced from about 200 μm to 100 μm, and from about 800 μm to 200 μm, respectively. The average micro hardness of the primary carbides increased from HV 1409 ± 44 Kgf/mm^2^ to HV 1493 ± 32 Kgf/mm^2^.(2)ECP did not induce changes in the crystal type of the primary carbides. The crystal structures were both orthogonal. However, the crystal lattice parameters a, b, and c decreased by 0.005 Å, 0.012 Å and 0.005 Å, respectively, because the average content of Cr in the carbides decreased.(3)The main reasons for carbide refinement are the distortion of the external electron layer of crystal embryos and the differences in energy between Fe ions and Cr ions caused by ECP. Refinement of the primary carbides caused the average content of Cr in the carbides to decrease, and caused the Cr/Fe ratio from the center to the edge of the carbide particles to decrease.(4)There were more twins and antiphase boundaries on the (001) plane of the primary carbides and more mixed dislocations in the substructure after modification of the HHCCI melt by ECP. Due to an increase in defect density, the average micro hardness of the primary carbides increased.

## Figures and Tables

**Figure 1 materials-12-00032-f001:**
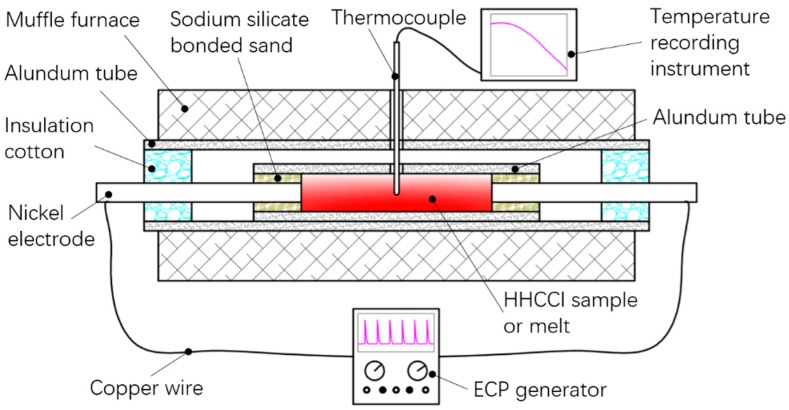
Schematic of the experimental device, where HHCCI is hypereutectic high chromium cast iron and ECP is electric current pulse.

**Figure 2 materials-12-00032-f002:**
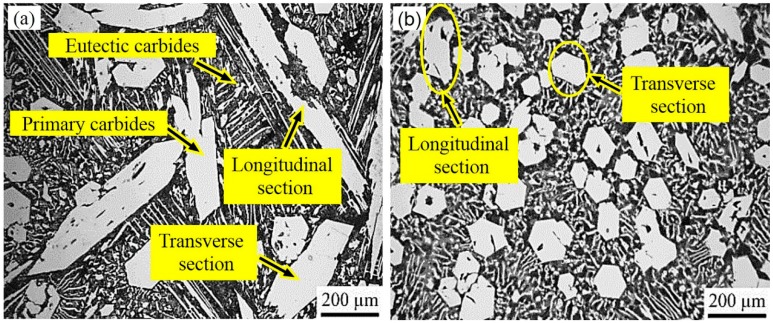
Morphology of the primary carbides (samples were etched with 5% FeCl_3_ aqueous solution): (**a**) non-ECP sample; (**b**) ECP sample.

**Figure 3 materials-12-00032-f003:**
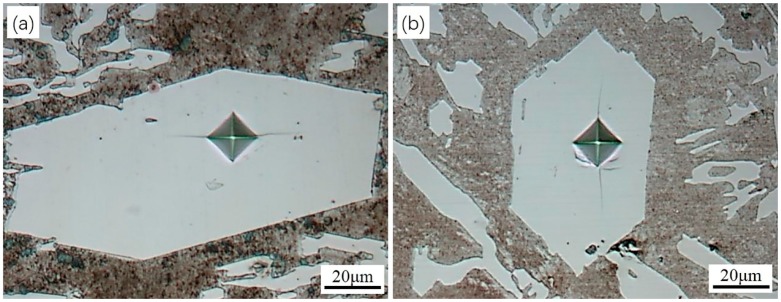
Indented transverse primary carbides: (**a**) non-ECP sample; (**b**) ECP sample. The load was 300 gf and the time of holding stage was 10 s at the load peak.

**Figure 4 materials-12-00032-f004:**
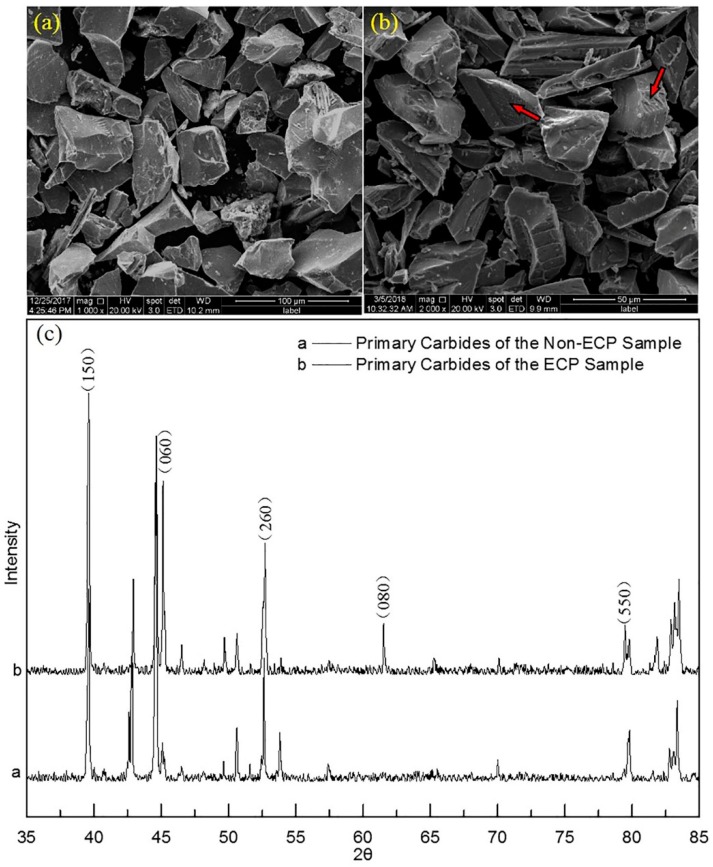
(**a**) Scanning electron microscopy (SEM) image of primary carbide powders for the non-ECP sample; (**b**) SEM image of primary carbide powders for the ECP sample; (**c**) corresponding X-ray diffraction (XRD) pattern.

**Figure 5 materials-12-00032-f005:**
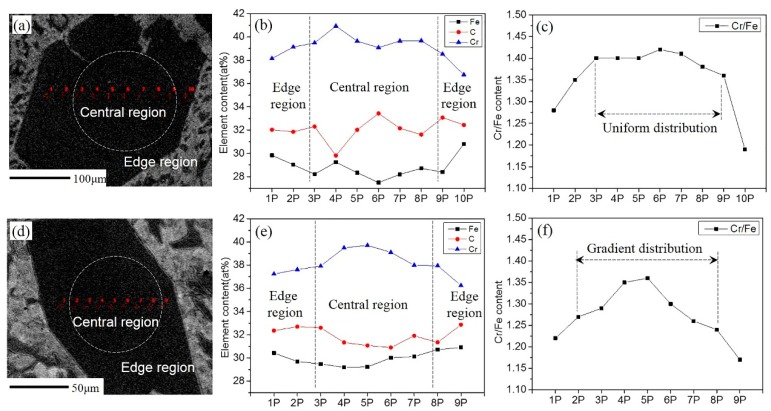
Contents and distribution of elements in the transverse sections of primary carbides. For the non-ECP sample: (**a**) electron probe micro-analyzer (EPMA) positions, (**b**) the distribution of elements, and (**c**) the trend of Cr/Fe atomic ratios. For the ECP sample: (**d**) EPMA positions, (**e**) the distribution of elements, and (**f**) the trend of Cr/Fe atomic ratios.

**Figure 6 materials-12-00032-f006:**
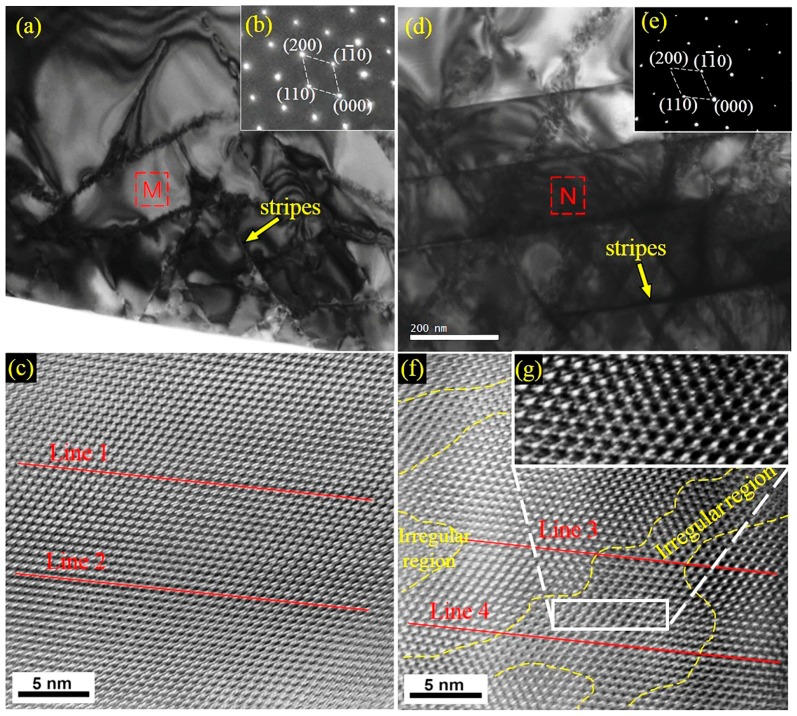
(**a**) Bright-field transmission electron microscopy (TEM) micrograph of primary carbides in the non-ECP sample, where area M is the position for high resolution TEM (HRTEM); (**b**) corresponding selected area diffraction patterns (SADP); and (**c**) corresponding HRTEM. (**d**) Bright-field TEM micrograph of primary carbides in the ECP sample, where area N is the position for HRTEM; (**e**) corresponding SADP; (**f**) corresponding HRTEM; and (**g**) corresponding magnified region.

**Figure 7 materials-12-00032-f007:**
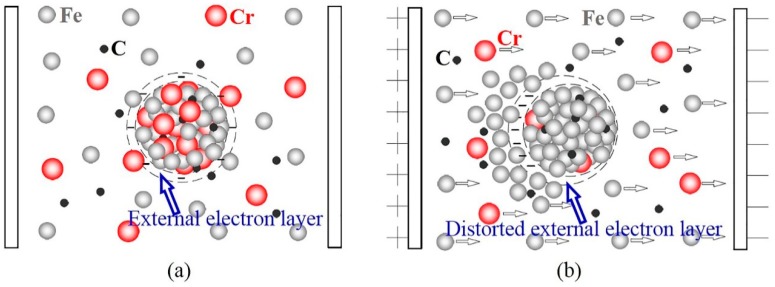
Schematic of the crystal embryo growth of primary carbides in the two samples: (**a**) non-ECP sample; (**b**) ECP sample.

**Figure 8 materials-12-00032-f008:**
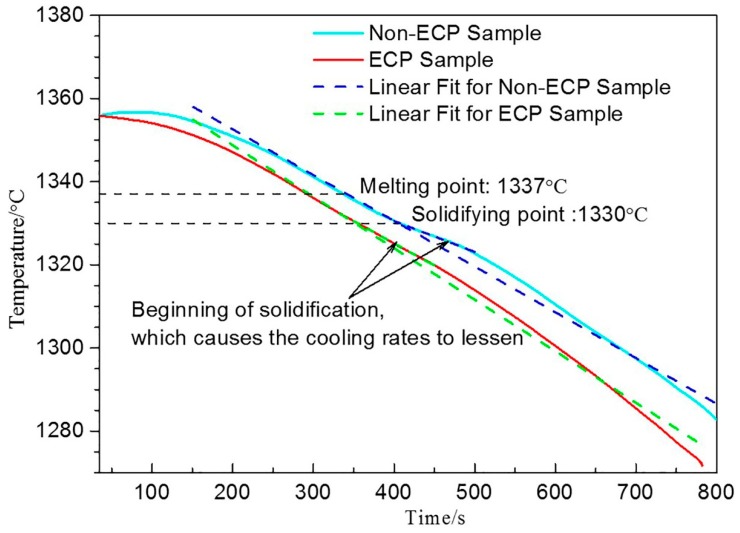
Cooling curve of the HHCCI during solidification.

**Figure 9 materials-12-00032-f009:**
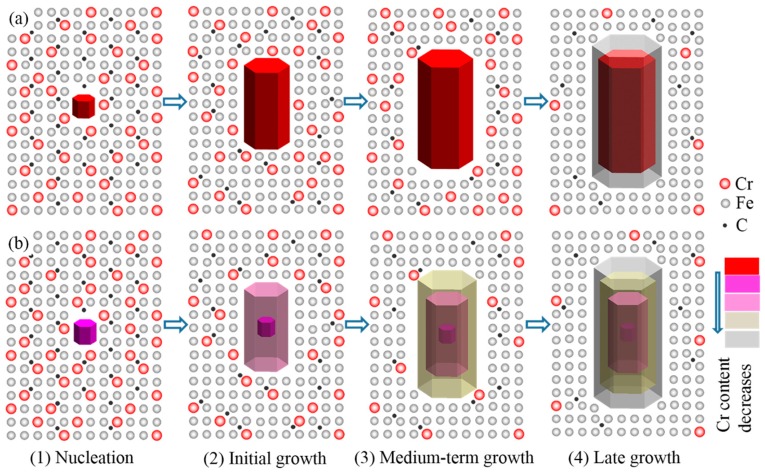
Schematic of the growth of carbides: (**a**) for the non-ECP sample; (**b**) for the ECP sample.

**Table 1 materials-12-00032-t001:** Average contents of elements in primary carbides in the HHCCI.

	Fe/(at%)	C/(at%)	Cr/(at%)
Non-ECP Sample	28.82 ± 0.35	32.07 ± 0.12	39.11 ± 0.23
ECP Sample	29.97 ± 0.25	31.89 ± 0.07	38.14 ± 0.17

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
