# Peer review of "Change in Primary (Cr, Fe)7C3 Carbides Induced by Electric Current Pulse Modification of Hypereutectic High Chromium Cast Iron Melt"

_materials, 2018, doi:10.3390/ma12010032_

Round 1
Reviewer 1 Report
Dear Authors,
The publication is interesting. The possibility of refinement of carbide precipitations in high-chromium cast iron using the pulse method has been described. Both the information contained in the introduction and references to literature are sufficient. Many research methods have been used which significantly enrich the article.
However, some shortcomings have not been avoided. They should be explained before publishing.
Best regards,
Reviewer

Author Response
Dear Reviewer,
Thank you very much for your comments about our manuscript (Title: Change of Primary (Cr, Fe)7C3 Carbides Induced by Electric Current Pulse Modifying Melt of Hypereutectic High Chromium Cast Iron, No: materials-400187). We have studied the comments carefully and have made corrections which we hope meet with you approval. It should be noted that the revised texts have been highlighted in the red or blue colour (the texts revised according to recommendations of professional English editing company were highlighted in the blue colour). Please refer to the attachment for details.
kind regards,
Yours sincerely,
Baoyu Geng, Rongfeng Zhou, Lu Li, Haiyang Lv, Yongkun Li, Dan Bai, Yehua Jiang

Reviewer 2 Report
Paper is interesting although require improvements before publication. Some detailed remarks are listed below:
1. English Native is recommended before publication.
2. Authors stated that chemical composition differ for Non-ECP Samples and ECP samples. In fact, there are very little differences. What is the error level of your measurements? Please indicate in the work.
3. Figure 5 does not convince me in connection to their analysis. It seems that it could be an over-interpretation.
4. The cooling curves are the weakest point in this article. I do not see characteristic stops on these curves.
Author Response

(The authors gave the same response as above.)

Reviewer 3 Report
The paper deals with the refinement of the chromium primary carbides in hypereutectic high chromium cast iron by using the electric current pulse. In the experiment, numerous testing methods were used from x-ray diffraction analysis to HRTEM. Therefore, a large number of results can be found in the paper.
The paper also contains an extensive introduction and a discussion of the results. The discussion is really very comprehensive and maybe even too long. It also contains a comparison to others authors as well. The number of references is also sufficient and up-to-date.
Recommendation for authors.
1) A large part of the discussion is devoted to the influence of ECP on the distortion of the external electrical layer. Yet, this topic is not mentioned in either the abstract or the introduction.
2) How long was the heating of the samples to 1355 °C? I suppose the soaking time was a time of holding at the temperature and not the whole heating.
3) How did the sample cool down? Was the effect of the cooling rate taken into account?
4) The labels in some of the figures are poorly readable, as for example arrows in Figure 2 or lines in Figure 5.
5) Was the measurement of the size of primary chromium carbides done in the transverse section? What would the authors expect in the longitudinal direction?
6) Why was the XRD analysis performed on ground samples and not on the full material? What was the particle size after grounding and how was the grounding done? Was the internal structure affected during milling, for example by increasing the temperature?
7) The experiment contains one set of ECP parameters. How have these parameters been chosen and what can be expected from other parameters?
8) The Cr/Fe content in Figure 4 c and d should be without units because it is a ratio.
Author Response

(The authors gave the same response as above.)

Reviewer 4 Report
In my opinion it would be appropriate to add the standard deviation in the Table 1
line 156: ....right-field TEM... correctly: ....bright-field TEM
line 271: ...(see Fig.3e and f)...correctly: (see Fig.4e and f)
line 278: ...(see Fig.3b and c)...correctly: (see Fig.4b and c)
line 296: ...(the red arrow in Fig.3(b))...: (the red arrow in Fig.4(b))
Author Response

(The authors gave the same response as above.)
